# Visual Analysis of Vessel Behaviour Based on Trajectory Data: A Case Study of the Yangtze River Estuary

Ye Li  and Hongxiang Ren *

Navigation College, Dalian Maritime University, Dalian 116026, China; liye@dlmu.edu.cn
* Correspondence: dmu_rhx@dlmu.edu.cn; Tel.: +86-1894-092-9166

**Abstract:** The widespread of shipborne Automatic Identification System (AIS) equipment will continue to produce a large amount of spatiotemporal trajectory data. In order to explore and understand the hidden behaviour patterns in the data, an interactive visual analysis method combining multiple views is proposed. The method mainly includes four parts: using a trajectory compression algorithm that takes into account the vessel motion characteristics to preprocess the vessel trajectory data; displaying and replaying vessel trajectories based on Electronic Chart System (ECS), and proposing a detection algorithm for vessel stay points based on the principle of spatiotemporal density to semantically label vessel trajectories; using the Fast Dynamic Time Warping (FastDTW) similarity measurement algorithm and the Ordering Points to Identify the Clustering Structure (OPTICS) clustering algorithm to cluster vessel trajectories to show the differences and similarities between vessel traffic flows; and showing the distribution of vessels and the variation trend of vessel density based on the vessel heatmap. Based on the AIS data of the Yangtze River Estuary, three cases are used to prove the usefulness and effectiveness of the system in vessel behaviour analysis.

**Keywords:** Automatic Identification System (AIS); vessel trajectories; vessel behaviour; visual analytics

## 1. Introduction

Vessel trajectory data mainly refers to the Automatic Identification System (AIS) data. AIS is an advanced radio technology that combines Global Positioning System (GPS), very high frequency (VHF), and data processing technology to enable different marine entities to exchange relevant information in a strictly defined format. This may be a simple exchange of position, course, speed and identity information between individual vessels, or a more complex data exchange between professional shore equipment and buoy positioning equipment [1,2]. AIS plays a significant role in ship collision avoidance, information exchange, maritime surveillance, etc. Its correct use helps to strengthen the safety of life at sea, improve the safety and efficiency of navigation, and protect the marine environment [3].

AIS data has temporal and spatial information, and the amount of information is very large. In only the lower reaches of the Yangtze River, nearly 100 million pieces of AIS data are generated every day, which typical of spatiotemporal big data. According to the characteristics of AIS spatiotemporal big data, through visualization and analysis, AIS can track vessel trajectories and statistics and analyze vessel traffic flows, cargo flows, etc. This has extremely important significance for guiding ship operation, maritime surveillance, and shipping trade [4]. At present, the analysis and research of AIS big data has gradually become a hot spot and challenge in the field of maritime research [5–8]. Huang et al. [9] mainly studied the compression and visualization of large vessel trajectories and their graphics processing unit (GPU) acceleration implementation. Zhou et al. [10] proposed a ship classification method based on behaviour clustering by analyzing port AIS data. Wang et al. [11] proposed a system identification modelling method for ship maneuvering motion based on support-vector machines (SVM). These studies either focus on how

to clearly display the AIS trajectory data or focus on analyzing the vessel behavioural characteristics behind the data. Although certain results have been achieved, the two are not organically unified.

Therefore, this paper proposes a visual analysis method combining multiple views to explore and analyze vessel behaviour patterns. The method in this paper is mainly divided into four parts: preprocessing the vessel trajectory data based on a trajectory data compression algorithm that takes into account the characteristics of vessel motion; displaying and replaying vessel trajectories based on the Electronic Chart System (ECS), proposing a detection algorithm for vessel stay points based on the principle of spatiotemporal density to semantically label vessel trajectories; using the fast dynamic time warping (FastDTW) similarity measurement algorithm [12] and the ordering points to identify the clustering structure (OPTICS) density clustering algorithm [13,14] to cluster vessel trajectories to show the differences and similarities between vessel traffic flows; displaying the vessel heatmap based on the principle of kernel density, and intuitively showing the trend of vessel distribution and density changes.

The rest of this paper is organized as follows. In Section 1, related work is discussed. In Section 2, an overview of the system is given. In Section 3, the data and proposed methodology are explained in detail. In Section 4, visual components and interactive tools for vessel behaviour analysis are presented. In Section 5, a case study of the Yangtze River Estuary is conducted, and results are analyzed. Finally, Section 6 concludes the paper with recommendations for further research.

## 2. Related Work

### 2.1. Visualization of Trajectory Data

With the widespread use of shipborne AIS equipment, there are more and more methods for statistical analysis of large-scale AIS data. Because vessel trajectory data is typical spatiotemporal big data, which contains complex temporal and geographic information, efficient visualization tools and methods are needed to help people analyze trajectory data.

Willems et al. [15,16] and Scheepens et al. [17] proposed the use of density maps to interactively explore the attributes of vessel trajectory data and analyze vessel motion. The experimental results showed that they are effective in detecting abnormal vessels. However, the method has a high requirement on hardware performance. Wang et al. [18] proposed an interactive trajectory visualization analysis tool and proved its effectiveness in detecting vessel motion patterns and potential illegal activities and pointed out that the tool's visual analytics environment is relatively simple and needs to be improved. Wu et al. [19] constructed a global shipping density map based on AIS data, showing the distribution of global vessels and traffic. However, the parameter setting of the method depends on personal experience and lacks interpretability. Jin et al. [20] proposed a visual analytics framework to interactively explore the characteristics of vessel behaviour by means of integrating visualization with data mining and a human–computer-interaction-controlling model. However, since the framework cannot support analysis of vessel behaviour with big data sets in real time, its efficiency must be improved. Zou et al. [21] proposed a 4D time density algorithm, applied volume visualization techniques to visualize the resulting density volume, and explored the movement patterns of moving objects in space and time through a real application case. However, this algorithm has the limitations of high computational complexity and low rendering efficiency. He et al. [22] assessed and explored AIS data quality visually, using a visual analysis platform for AIS data quality. Zhang et al. [23] proposed an interactive visual method to identify trajectory stay points, but the method is only effective over a short period of time, and the effect is poor when the trajectory is long. Storm-Furru et al. [24] designed an interactive web-based geographic visual analysis tool in order to use the trajectory of a fishing boat to identify unreported catch operations. Oztürk et al. [25] developed a tangible visual analysis tool that uses AIS data to analyze maritime traffic on a spatiotemporal basis. This approach could analyze the

safety structure of fairways and individual vessels at a micro level and is a good attempt at visualization technology in the field of maritime traffic.

### 2.2. Data Mining with Trajectory

Trajectory data is a representation of a vessel's navigation behaviour, which can map out information such as the vessel's navigation mode and behaviour rules. At present, many scholars have conducted in-depth and effective discussions on the research of vessel trajectory data mining methods.

Wei et al. [26] designed an AIS trajectory compression algorithm considering space and motion features of trajectories based on the Douglas–Peucker and sliding window algorithm, which can compress AIS trajectories according to ships' behavioural characteristics. The main drawbacks of the algorithm are that the threshold coefficient cannot be determined adaptively and that the running time is long. Chen et al. [27] proposed an improved SRC (sparse representation classification) model based on Lp-norm to classify vessel motion patterns, which can effectively classify vessel motion patterns in inland waterways. The proposed model is superior to those of other representative classification methods. Zhao et al. [28] proposed a combined method composed of DBSCAN (Density-based spatial clustering of applications with noise) algorithm and recurrent neural network, which can quickly detect abnormal behaviour of vessels in speed, course, and route. However, the method has a high requirement on the quality of sample vessel trajectory data, which makes practical application more complicated. Liu et al. [29] proposed a real-time identification framework of regional collision risk based on AIS data, which provides the possibility for collision risk monitoring and collision risk analysis. Liu et al. [30] took inbound and outbound ships as the research objects and, based on the traffic flow structure characteristics driven by AIS trajectory data, proposed a method to estimate the navigation capacity of busy waterways. The proposed approach provides support for the design of channels and the determination of scheduling schemes for ships in busy waterways. Chen et al. [31] proposed a convolutional neural network-ship motion pattern classification (CNN-SMMC) algorithm based on AIS data, which can effectively extract detailed ship movement information and fully realize ship motion classification. Similar to the CNN classification method, this method also has the limitations of difficult parameter setting and high requirements for sample quality. Wang et al. [32] combined ship AIS and channel geocoding data to model traffic flow characteristics, estimated the correlation between ship speed and congestion, and analyzed the shipping traffic performance of different segments. Their work provides useful insights into testing the rationality of speed limits in other waterways or shipping channels. Yan et al. [33] combined semantic trajectory and graph theory to realize the extraction and expression of maritime traffic routes in large sea areas in a relatively simple and easy way. However, their method does not consider the influence of ship types, so the robustness of the method needs to be strengthened.

Although many valuable conclusions have been obtained in the previous work, the visualization method focuses on how to display AIS trajectory data efficiently and with high quality, while the vessel behaviour analysis focuses on the efficiency of algorithms and ignores visualization. At present, there is still a lack of an interactive visualization system for comprehensive and efficient analysis of vessel behaviour. Therefore, this paper proposes more systematic and comprehensive visual analysis tasks by combining vessel trajectory data visualization and vessel-behaviour-oriented feature visualization and designs and implements a visualization system with high efficiency, concise, multi-view cooperation, which intuitively displays vessel behaviour and traffic flow characteristics.

### 3. System Overview

This paper designs a visual analysis system to explore vessel behaviour and statistically analyze vessel traffic flow. Figure 1 shows the workflow of the system and mainly includes three parts: a data processing module, a visual component module, and a visual interaction module.

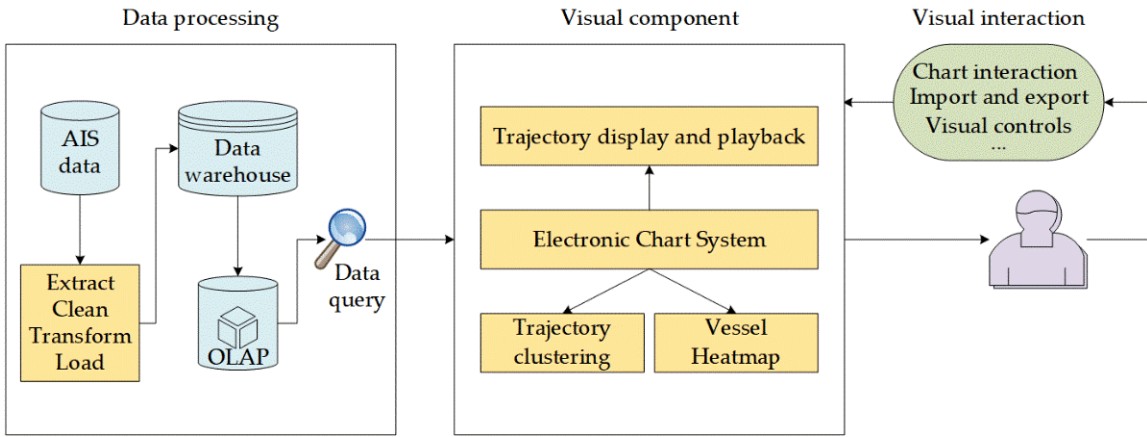

**Figure 1.** Overview of system workflow.

(1) The data processing module mainly realizes data preprocessing and storage. Due to the high sampling rate and multiple sources of AIS data, the amount of data is large and there is duplication. In order to facilitate storage and calculation, the original data need to be deduplicated and compressed. Due to equipment problems, AIS data is prone to noise and data missing (as shown in Figure 2), so it is necessary to denoise and repair the trajectory data. The visualization of large-scale trajectory data cannot be carried out by interactive visual analysis because of its large storage capacity and slow query acquisition. This paper uses data warehouse [34] to manage data, which can effectively reduce the calculation cost of the system and improve the work efficiency of the system.

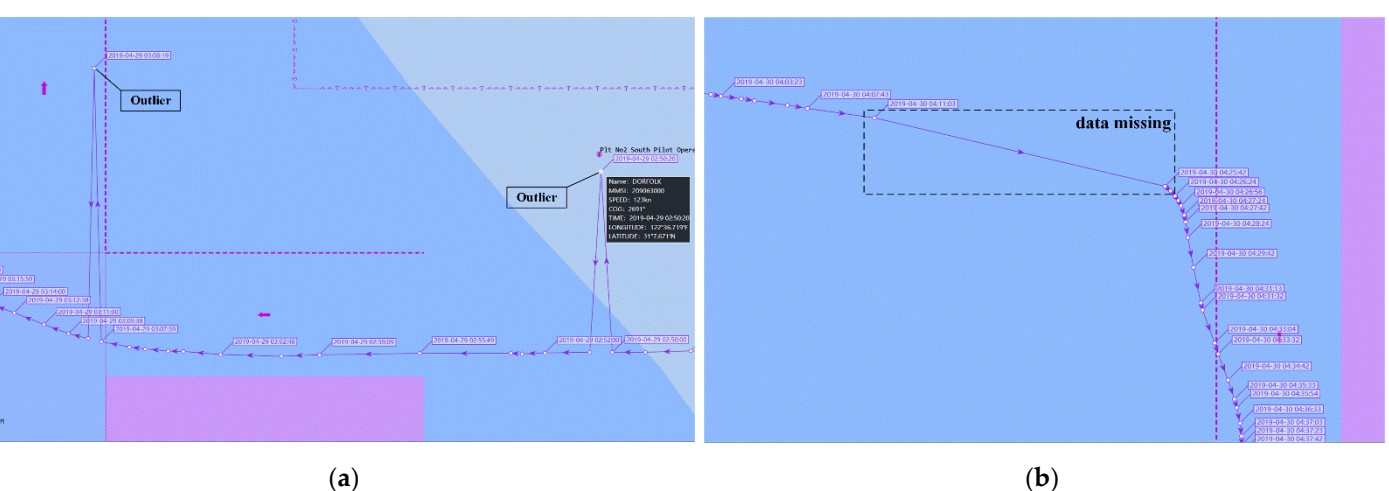

| (**a**) | (**b**) |

**Figure 2.** The trajectories with outliers recorded by AIS loggers: (**a**) an AIS trajectory with two noise points; (**b**) an AIS trajectory that suffers from the data missing problem.

(2) The visual component module mainly realizes the detection of vessel stay points, the display and playback of vessel trajectories based on ECS, the analysis of vessel traffic flow based on density clustering, and the vessel heatmap. Through collaborative work between components, it effectively helps users understand data distribution, overview trajectory data relationships, explore vessel behaviour patterns, and observe detailed trajectory data.

(3) The visual interaction module designs rich interface interactions. By setting up a visual menu, users can complete basic operation of ECS: trajectory data import and export, trajectory data compression, trajectory data display and playback in different periods and regions, trajectory clustering analysis, vessel heatmap, and so on.

## 4. Data and Methodology

### *4.1. Data*

#### 4.1.1. Data Description

AIS data contains 27 kinds of AIS messages, among which position report, static report, and voyage report contain the most important vessel information. An AIS position report can generate an AIS trajectory point, while a vessel's AIS trajectory is a series of continuous trajectory points of the same vessel. The AIS position report is broadcast at an interval of 2–180 s, depending on the navigation status of the vessel and the type of AIS (Class A or Class B). According to the requirements of the International Telecommunication Union (ITU), the reporting intervals and further details for AIS are shown in Table 1 [35]. Therefore, AIS trajectory data can restore the movement of vessels to a certain extent. In addition to providing vessel position information, AIS data can also provide information such as destination, vessel type, size, and the current speed and course of the vessel corresponding to the trajectory point [4]. Therefore, correct use of AIS is helpful for judging ship movement and coordinating collision avoidance actions and plays an important role in preventing collisions, especially for ships navigating in waters with poor visibility. At the same time, it must also be noted that not every vessel is equipped with AIS; although some vessels are equipped with AIS, the use and settings are not standardized. AIS data may also be delayed, so the received data may not be correct. The main function of AIS data is ship identification and academic research, not real-time ship collision avoidance.

**Table 1.** Reporting interval requirements for AIS equipment.

| Class A Equipment | | Equipment Other than Class A | |
|---|---|---|---|
| **Ship's Dynamic Conditions** | **Reporting Interval** | **Platform's Conditions** | **Reporting Interval** |
| Ship at anchor or moored and not moving faster than 3 knots | 3 min | Class B "SO" [1] ship not moving faster than 2 knots | 3 min |
| Ship at anchor or moored and moving faster than 3 knots | 10 s | Class B "SO" [1] ship with a speed of between 2–14 knots | 30 s |
| Ship with a speed of between 0–14 knots | 10 s | Class B "SO" [1] ship with a speed of between 14–23 knots | 15 s |
| Ship with a speed of between 0–14 knots and changing course | 10/3 s | Class B "SO" [1] ship with a speed of greater than 23 knots | 5 s |
| Ship with a speed of between 14–23 knots | 6 s | Class B "CS" [2] ship not moving faster than 2 knots | 3 min |
| Ship with a speed of between 14–23 knots and changing course | 2 s | Class B "CS" [2] ship moving faster than 2 knots | 30 s |
| Ship with a speed of greater than 23 knots | 2 s | Search and rescue aircraft, AIS base station | 10 s |
| Ship with a speed of greater than 23 knots and changing course | 2 s | Aids to navigation | 3 min |

[1] Class B "SO" means an AIS equipment using Self-Organized Time-division Multiple Access (SOTDMA) technology;
[2] Class B "CS" means an AIS equipment using Carrier-Sense Time Division Multiple Access (CSTDMA) technology.

As a useful big data research object, AIS data has high research value. Through the visual analysis of AIS data, the potential pattern information of vessels can be obtained, and maneuvering behaviours and navigation characteristics of vessels can be identified.

#### 4.1.2. Data Preprocessing

Low-quality data will lead to low-quality mining results. To improve data quality, we must preprocess the data. In the first step, AIS data have high sampling rates and heterogeneity, which inevitably lead to noise, redundancy, and repetition, so it is necessary to clean the data. AIS data cleaning mainly includes deduplication, filling missing values,

and smoothing noise [36]. The second step is trajectory compression. Due to the high sampling rate of AIS equipment, the amount of trajectory data is large, which will seriously affect the storage and analysis efficiency. Therefore, it is necessary to compress the trajectory data. In this paper, a vessel trajectory compression method considering time and speed characteristics is adopted, and the specific method will be described in Section 4.2.1. In the third step, only the position report is insufficient in the trajectory data analysis, and the static report and the voyage report are indispensable. Therefore, it is necessary to integrate different message data together, that is, to perform data integration. In this paper, different messages are matched and integrated according to the common attributes of their Maritime Mobile Service Identities (MMSI) and the timestamps of messages. In the fourth step, it is necessary to perform data reduction because the integrated AIS data set is large and direct analysis without preprocessing will make such analysis impractical or unfeasible. This paper analyzes the specific tasks in detail; selects different attribute subsets for different task data sets; and removes irrelevant, weakly related, or redundant attributes.

### 4.1.3. Data Storage and Access

In order to facilitate data storage and acquisition, this paper uses a topic-oriented, integrated, and relatively stable data collection method, data warehouse, to manage data. The data warehouse in this paper adopts the traditional three-tier system structure. The bottom layer is the warehouse database server, which is a relational database system used for data extraction, cleaning, loading, and refreshing [37,38]; the middle layer is an online analytical processing (OLAP) server, which is responsible for multi-dimensional analysis such as the slicing, cutting, and drilling of data; the top layer is the customer layer, including query and analysis tools. The data warehouse model adopted in this paper is a virtual warehouse from the structural point of view, which is a collection of views on the operation database. For example, this paper stores AIS data from different port waters and different periods in view mode.

### *4.2. Methodology*

### 4.2.1. Trajectory Compression

At present, most vessel trajectory compression algorithms, such as Douglas–Peucker [39], Visvalingam–Whyatt [40], are based on the geometric features of trajectory. However, besides geometric features, trajectory data also contains motion features such as speed and time interval, and the motion state of the trajectory subject has great influence on the geometric shape of trajectory data. To solve this problem, in order to keep more characteristic information of the trajectory data and ensure the integrity of trajectory information better, this paper uses a trajectory data compression algorithm which considers the characteristics of vessel motion [41]. The algorithm sets the anchor point, then gradually opens the "window". At each step, there are two halting conditions—one based on the synchronous distance measurement and the other based on the difference in speed between previous and next trajectory segments. These speeds are not the measured speeds of the vessel because the algorithm assumes that these speeds are unavailable; instead, they are speed values derived from timestamps and locations. The specific simplification process is summarized in Algorithm 1:

---

**Algorithm 1: simplification algorithm considering vessel behavioural characteristics (SBC)**

---

Input: trajectory points list, $s[p_1, \cdots, p_n]$ (where trajectory point $p$ includes: timestamp, location);
distance difference threshold, $max\_dist\_threshold$;
speed difference threshold, $max\_spd\_threshold$.
Output: simplified trajectory points list
// the number of trajectory points in $s$
if $len(s) \leq 2$ then
return $s$
else
// initially set the halting condition to false
$is\_halt \leftarrow false$
$e \leftarrow 1$
while ($e \leq len(s)$ and not $is\_halt$) do
$i \leftarrow 1$
while ($i < e$ and not $is\_halt$) do
// $t$ representatives timestamp, $\Delta e$ and $\Delta i$ are timestamp difference
$\Delta e \leftarrow s[e]_t - s[0]_t$
$\Delta i \leftarrow s[i]_t - s[0]_t$
// $loc$ representatives the location of trajectory point
// $(x, y)$ representatives the (longitude, latitude) coordinates of a trajectory point
$(x'_i, y'_i) \leftarrow s[0]_{loc} + (s[e]_{loc} - s[0]_{loc})\Delta i / \Delta e$
// $dist$ is a function that takes two points and returns the distance between them
// $v$ representatives average speed between the previous and next trajectory segment
$v_{i-1} \leftarrow dist(s[i]_{loc}, s[i-1]_{loc}) / (s[i]_t - s[i-1]_t)$
$v_i \leftarrow dist(s[i+1]_{loc}, s[i]_{loc}) / (s[i+1]_t - s[i]_t)$
If $dist(s[i]_{loc}, (x'_i, y'_i)) > max\_dist\_threshold$ or $\|v_i - v_{i-1}\| > max\_spd\_threshold$ then
$is\_halt \leftarrow true$
else $i \leftarrow i + 1$
end if
end while
if $is\_halt$ then
// concatenates two trajectories
return $[s[0]] + SBC(s[i, len(s)], max\_dist\_threshold, max\_spd\_threshold)$
end if
$e \leftarrow e + 1$
end while
if not $is\_halt$ then
// a new sequence consisting of the first and last point in $s$
return $[s[0], s[len(s)]]$
end if
end if

---

**Step 1:** Enter the sequence points of a vessel trajectory $s$ (trajectory point includes: timestamp, location), distance threshold $max\_dist\_threshold$, and speed threshold $max\_spd\_threshold$. If the length of $s$ is greater than 2, initialize the halting condition $is\_halt$ to false and proceed to Step 2, otherwise return the sequence $s$.

**Step 2:** Starting from the second point, each point in $s$ is used as the anchor point, and the external forward traversal is performed to the end of $s$. If the anchor point does not reach the end of $s$ and $is\_halt$ is false, then proceed to Step 3; otherwise, proceed to Step 5.

**Step 3:** Let the subsequence before the anchor point be the moving point set $f$ of the anchor point. In $f$, starting from the second point, each point is the moving point of the anchor point, and the internal forward traversal is performed to the anchor point. In the internal traversal process, two halting conditions are verified: one is that the synchronous distance measurement (the spherical distance between the predicted position and the actual position of the moving point, and the predicted position of the moving point is obtained by Equations (1)–(4)) is greater than $max\_dist\_threshold$, and the other is that the difference in average speed between the previous and next trajectory segment separated by the moving

point is greater than *max_spd_threshold*. If the halting condition does not hold, shift the moving point forward to continue the internal traversal (if the moving point reaches the anchor point, stop the internal traversal, move the anchor point forward, and proceed to Step 2), otherwise *is_halt* is true and proceed to Step 4.

$$\triangle e = t_e - t_s \tag{1}$$

$$\triangle i = t_i - t_s \tag{2}$$

$$x'_i = x_s + (x_e - x_s) * \triangle i / \triangle e \tag{3}$$

$$y'_i = y_s + (y_e - y_s) * \triangle i / \triangle e \tag{4}$$

where $\triangle e$ is the timestamp difference between the anchor point and the first point, $\triangle i$ is the timestamp difference between the moving point and the first point, $(x_e, y_e)$ is the position of the anchor point, $(x_s, y_s)$ is the position of the first point, and $(x'_i, y'_i)$ is the predicted position of the moving point.

**Step 4:** Intercept the points after the moving point in $s$ to obtain a new sequence $s'$, and take $s'$ as the input for Step 1, perform recursive processing. Connect the first point in $s$ and the return value of the recursive function to form a new sequence and return it.

**Step 5:** If *is_halt* is false, a new sequence consisting of the first and last point in $s$ is returned. Otherwise, no processing is done.

### 4.2.2. Stay Point Detection

Identifying and analyzing vessel stay points (berthing points, anchoring points) is an important part of trajectory analysis. It can not only provide necessary basic information for the safe navigation and berthing of vessels, but also provide important decision support for port planning and construction. However, the existing methods either do not consider the time continuity of trajectory points, or only consider the time continuity, resulting in insufficient recognition ability of stay points. Based on the literature [42], this paper presents a density-based identification method of vessel stay points which considers the spatiotemporal aggregation of trajectory points and takes into account the time continuity and directivity of trajectory points. The algorithm mainly includes three steps: density calculation, stay point identification, and type judgment. Figure 3 shows the framework of stay point detection algorithm.

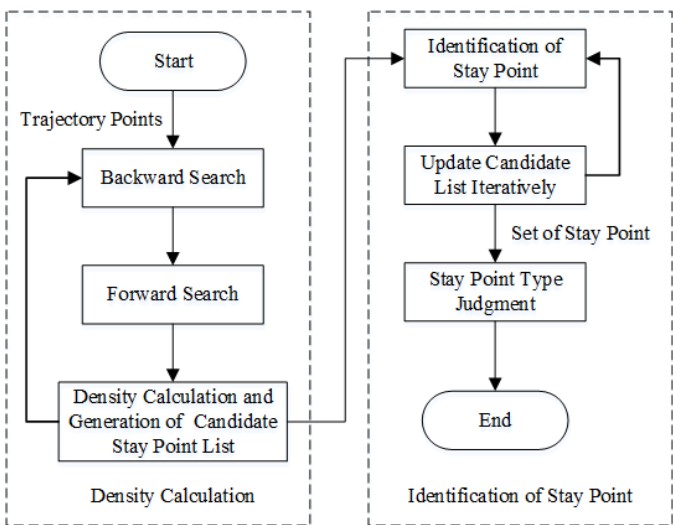

**Figure 3.** Algorithm framework of stay point detection.

**Step 1:** Density calculation. In this step, each point of the trajectory point sequence is taken as an anchor point in turn. According to the distance threshold, by searching

backward and forward along the timeline, the density interval of each anchor point is obtained (there is a longest continuous subsequence in a given vessel trajectory point sequence, and if the spherical distance between any point in the subsequence and the anchor point is less than the given distance threshold, the subsequence is the density interval of the anchor point). Based on the density interval, the density (number of trajectory points in density interval) and the time span (time difference between the starting point and the ending point of the density interval of trajectory points) of the anchor point are calculated. If the time span of the anchor point is longer than the time threshold, the anchor point is a candidate stay point, and a candidate stay point list (hereinafter referred to as CSPs) is generated.

**Step 2:** Identification of stay points. The candidate stay points in the CSPs have met the distance threshold and the time threshold. However, the density intervals of these candidate stay points may overlap, so in the process of stay point identification, the CSPs are iteratively identified and updated according to density from high to low, and disjoint stay points are obtained until the CSPs is empty.

**Step 3:** Type judgment. There are two states of vessel stay: berthing and anchoring. During anchoring, the vessel will show periodic yaw due to the influence of hydrodynamic force, wind force, and anchor chain tension. Especially when the vessel is anchored in tidal waters, the position of the vessel will change with the change of the tidal stream. Therefore, the standard deviation of the positions and speeds of the stay point is larger during the anchoring period and smaller during the berthing period. Calculate the standard deviation of the positions and speeds at each stay point. If the standard deviation of the stay point is greater than the threshold, it is an anchoring point. Otherwise, it is a berthing point.

### 4.2.3. Trajectory Clustering

Clustering analysis divides data objects into clusters according to data similarity and keeps the maximum distance between clusters and the minimum distance within cluster. Vessel trajectory clustering is an extension of clustering analysis in spatiotemporal trajectory. Its purpose is to group vessel trajectories with similar behaviours into one class based on spatial or temporal similarity. Through clustering, we can find vessel navigation patterns, analyze navigation rules, detect abnormal behaviours, and predict future motion behaviours. There are two necessary parts in the process of vessel trajectory clustering. One is similarity measurement between trajectories, and the other is trajectory clustering.

(1) Similarity measurement. Trajectory similarity is usually calculated by a distance function. In this paper, the FastDTW distance measure is used to express the similarity between two trajectories. FastDTW has the same advantage as DTW, which is that it is not limited by whether the number of points between two trajectories is the same, but it greatly reduces the calculation complexity of DTW and improves the clustering efficiency. FastDTW combines restriction and data abstraction to speed up the calculation of DTW.

(2) Trajectory clustering. In this paper, the density-based clustering algorithm OPTICS is used to cluster vessel trajectories. The OPTICS algorithm is chosen because it is specially developed for processing spatial data. It can obtain clusters of arbitrary density without setting the number of clusters as an explicit parameter. Perhaps most importantly, it can handle clusters of different densities commonly found in maritime traffic.

In this paper, the FastDTW similarity measurement algorithm and the OPTICS clustering method are combined to realize fast vessel trajectory clustering without parameters.

### 4.2.4. Heatmap

Vessel density refers to the number of vessels in a certain instantaneous unit area of water. It reflects the density of vessels in the waters, and also reflects the busy degree and dangerous degree of vessel traffic in the waters to a certain extent. It plays an important role in vessel traffic diversion, anchorage range planning, ship routing system design, and so on. The traditional vessel density representation method is the vessel density distribution map. Although it can qualitatively and quantitatively reflect the spatial distribution, it is

not convenient and intuitive, and it is difficult to highlight key areas, which makes the user experience poor. Heatmaps can simply aggregate data using different colors and brightness to reflect different density degrees, and intuitively show the density degree of spatial data [43]. Therefore, this paper proposes using a heatmap to represent vessel density distribution.

The heatmap is based on kernel density analysis [9,44]. In kernel density analysis, the number of vessels per unit area is distributed to the designated neighborhood with the change trend of kernel function, and the overlapping areas are added. Finally, the continuous and smooth vessel density in various places in the water area is obtained, which intuitively reflects the distribution of discrete measured values in the continuous area. The following is a brief production process of a heatmap:

**Step 1:** Vessel unit density calculation. The water area is rasterized, and the number of vessels in each instantaneous unit area of the water area is counted, and the vessel density distribution map is obtained.

**Step 2:** Kernel density analysis. Gaussian function (Equation (5)) is selected as the kernel function to calculate Gaussian kernel. The weighted Gaussian kernels are weighted locally with the vessel unit density as the weight, the weighted Gaussian kernels are added, and the added results are normalized. Finally the kernel density analysis value is obtained.

$$G(x,y) = 1/2\pi\sigma^2 \cdot exp\left(-\left(x^2 + y^2\right)/2\sigma^2\right) \tag{5}$$

**Step 3:** Generate a bitmap, use the kernel density analysis value as an index, map colors from the preset gradient color band, and recolor the bitmap to realize the heatmap.

## 5. Visual Design

Trajectory visualization in this paper is developed by DirectX (a collection of application programming interfaces developed by Microsoft to handle the higher demands of graphic programming found in today's applications), which ensures the ability to present thousands of data points or lines in computer terminals and supports real-time fast drawing of a large number of trajectories [45]. All visual views are based on ECS, scaled and converted synchronously with ENCs (Electronic Navigational Charts). In order to improve visualization efficiency, this paper uses a trajectory compression algorithm which considers vessel motion characteristics (Section 4.2.1 of this paper) to compress trajectory data on multiple scales. In large-scale displays, more trajectory points should be reserved, and the trajectory should be compressed slightly by controlling the threshold value, and vice versa. Based on the demand of vessel AIS data analysis in the industry (for example, maritime surveillance, fleet dynamic monitoring, ship motion law analysis, and so on) [46] and the above data processing flow and analysis method, this paper designs a multi-view visual analysis system, which mainly includes the visual interaction and three visual views: trajectory display and playback, trajectory clustering, and ship heatmap.

### 5.1. Trajectory Display and Playback

Vessel trajectory display and playback have a wide range of application scenarios, which can be seen in the analysis of vessel navigation characteristics, the study of water traffic mechanism, the identification of violations, accident investigation and evidence collection, etc. According to the actual requirements of trajectory visualization, this paper designs two trajectory display and playback modes: multi-vessel mode and single vessel mode.

(1) Multi-vessel mode. AIS data records the trajectories of almost all vessels on the route, which is very suitable for research or forensics work such as vessel trajectory distribution, traffic flow analysis, accident investigation, and so on. However, the navigable waters of vessels are wide, and there is a large number of vessels. Only some areas and some vessels are relevant. If they are displayed and replayed directly without screening, it will affect the efficiency of data query and trajectory rendering and increase unnecessary

interference. Therefore, this paper develops multi-ship trajectory display and playback functions. By loading external files or setting the range of interest area (longitude and latitude of upper-left and lower-right vertices in a rectangular area) and time span (start and end time), all vessels sailing through interest area in the set time period are queried, and the loading or query results are presented in the form of list. The result list lists the vessels' names, MMSI, number of trajectory points, start times, end times, and other information. By clicking on the result list, the vessel trajectories can be selectively displayed and replayed.

(2) Single vessel mode. When we are only required to visualize the trajectory data of a certain vessel and conduct microscopic studies such as the analysis of vessel motion characteristics and navigation behaviours, the multi-vessel mode becomes less suitable. When the historical position of the vessel is unknown, it is impossible to query the trajectory through the region of interest. Therefore, this paper develops the function of the single vessel trajectory display and playback. By setting any one of the vessel name, MMSI, or call sign and time span of the particular vessel, the single vessel trajectory can be queried quickly, and the query results can be displayed or replayed. The single vessel trajectory mode can show the position, speed, course, and other microscopic information of the vessel in the chart. The proposed method of vessel stay point identification based on density (Section 4.2.2 of this paper) can effectively identify the berthing and anchoring states of the vessel and label the vessel trajectory semantically.

Both modes support the export function of query results and save them in CSV format. Users can choose whether to display or replay only, or both at the same time.

### 5.2. Trajectory Clustering

The purpose of designing a trajectory clustering view is to group trajectories with similar behaviours into one class based on spatial or temporal similarity and to separate outlier trajectories so as to find the movement patterns of vessels and analyze the movement law. In this paper, the FastDTW algorithm and the OPTICS clustering algorithm are used to achieve fast vessel trajectory clustering. Trajectory data can be obtained by default setting, manual parameter setting, or external data import, and the obtained data can be further screened and compressed by manual parameter setting. Trajectory clustering visualization is based on multi-vessel trajectory display, which supports fast visualization of clustering results. Different trajectory clusters are distinguished by color and synchronized with chart color mode switching. This view provides the function of exporting clustering results, which is convenient for follow-up research.

### 5.3. Vessel Heatmap

Although the trajectory map can show the distribution of trajectories intuitively, when showing more trajectory data, it will cause visual clutter, which makes it difficult to highlight key areas and makes the user experience poor. Therefore, this paper designs a vessel heatmap to show the trend of vessel distribution and density change from the perspective of time and space; color is used to reflect data, and progressive color is used to map the amount of data in consecutive time periods. The red area indicates a large number of vessels and a high density, while the green area represents a relatively small number of vessels and a relatively low density. Vessels are mobile; with the change of time, the spatial position and quantity of vessels in the waters will change, and the distribution and density of vessels in the waters will also change accordingly. By continuously updating the heatmap, the distribution and density change trend of vessels can be displayed intuitively. The heatmap of the selected area can be observed and updated by dragging the time progress bar.

### 5.4. Interaction Design

Vessel trajectory data contains a large amount of information, and it is difficult to show spatial and temporal information, relationship information, and statistical information

of trajectory data reasonably with a single visual scheme. Therefore, this paper adopts the method of multi-view linkage and cooperation and uses interactive means to support visual analysis. The following are some interactions supported by this system:

(1) Chart interaction. All visualization in this paper is based on ECS. The system supports basic display, background control, zoom, roaming, hierarchical display, and so on.

(2) Labels. In the trajectory display view, the trajectory point information is displayed by a pop-up window, and the user can control the display and hiding of the trajectory point pop-up window by moving the mouse; stay points are displayed by static icons. Vessel trajectory data granularity changes with the scale of ENCs; trajectory labels also change so as to prevent the overlapping and occlusion of labels.

(3) Visual controls. The user selects time threshold and range threshold through the input control in the toolbar to control the import and export of trajectory data; the trajectory playback speed and progress are controlled by a sliding bar control; the starting and stopping of trajectory display, trajectory playback, and heatmaps are controlled via button control.

## 6. Case Analysis

In this paper, the available AIS historical data collected from 1 to 30 April 2019 consisted of more than 6.5 million pieces of AIS information broadcast by about 10,000 different vessels in the Yangtze River Estuary regions, covering an area of more than 2000 square kilometers spanning from 122°10′ E to 122°40′ E in longitude and from 30°55′N to 31°15′N in latitude. After preprocessing the data, such as cleaning and compression, 1.5 million pieces of position report data were extracted.

For the three visual analysis modules proposed in Section 5, three cases are shown, corresponding to Sections 5.1–5.3, to illustrate the usefulness and effectiveness of this system.

### 6.1. Vessel Trajectory Display and Playback

ECS is widely used in vessel navigation, vessel traffic management, vessel dispatching, and so on. All the views in this paper are displayed on the basis of it. Figure 4 is a chart of the Yangtze River Estuary. The system supports basic display, background control, scaling, roaming, and multi-layer display of ENCs.

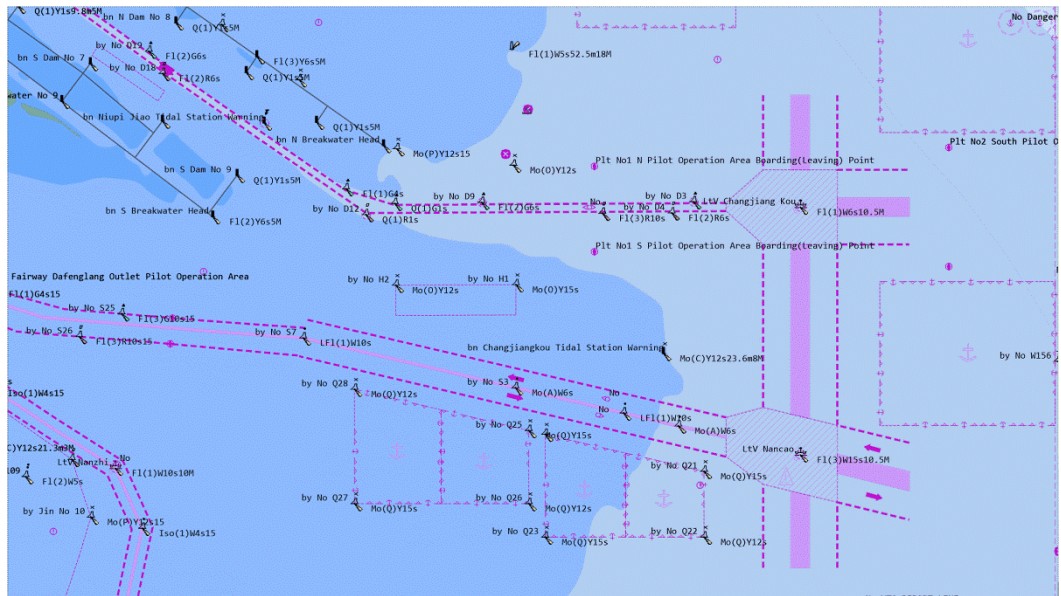

**Figure 4.** Chart of the Yangtze River Estuary.

Multi-vessel trajectory display is widely used in macroscopic traffic analysis fields such as vessel trajectory distribution and traffic flow analysis. Figure 5 is a multi-vessel trajectory display in which each green line represents a vessel historical trajectory. Figure 5a is the trajectory display from 1 to 3 April, and Figure 5b is the trajectory display from 1 to 10 April. The areas with concentrated lines in the Figure 5 are generally the waters such as customary vessel routes and anchorages. In the Figure 5, there are many vessels passing through area A, which is the customary route of some vessels. However, according to the routing system, area A is not a legal route. This finding has certain significance for future maritime surveillance and fairway setting. By comparing Figure 5a with Figure 5b, it can be seen that when the number of trajectories is large, visual clutter will be caused, and the number of trajectories should be reasonably controlled in actual use.

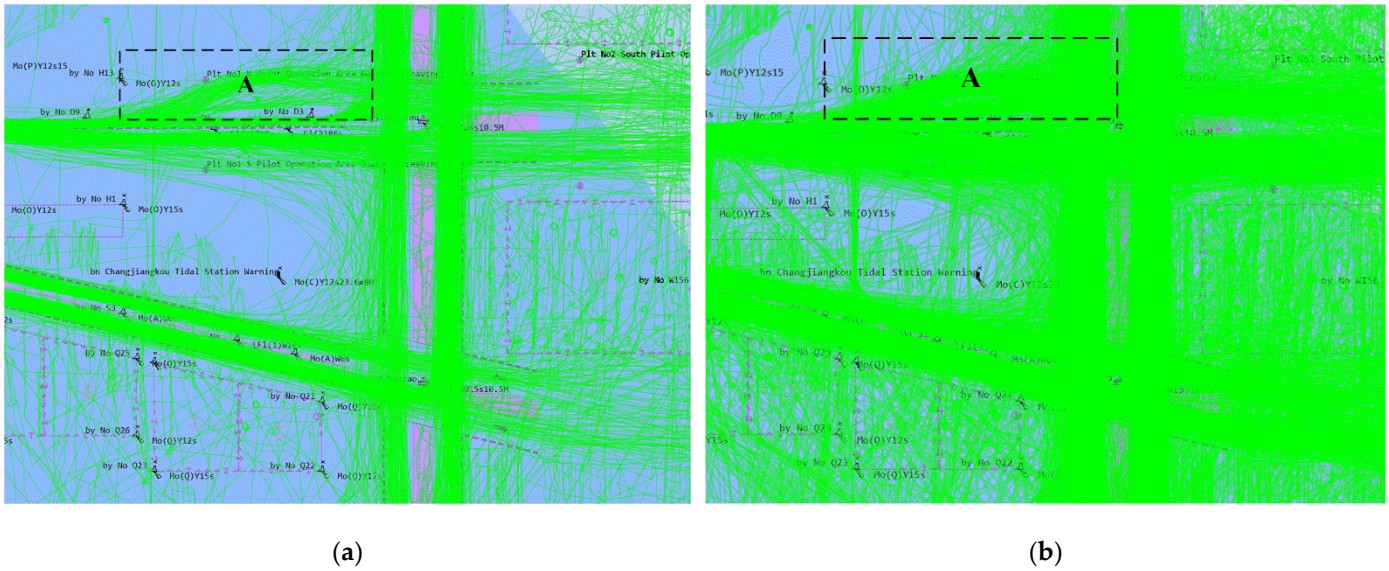

(**a**)  (**b**)

**Figure 5.** Multi-vessel trajectory display: (**a**) trajectory display from 1 to 3 April; (**b**) trajectory display from 1 to 10 April.

As shown in Figure 6, the historical trajectory of a vessel in the Yangtze River Estuary on 28 April 2019 is shown. In Figure 6a, the position points in the trajectory are represented by white dots with time labels, and the adjacent position points are connected by directed arrows. The red signs mark the stay points of the vessel. The details of the position point are displayed in the form of pop-up boxes, which are triggered when the mouse is placed over the position points. The granularity of vessel trajectory data changes with the scale of ENCs and is displayed hierarchically. As shown in Figure 6b, the historical speed of a vessel appears in the form of an interactive chart. The development of the single-vessel trajectory display function facilitates the microscopic research of vessel motion characteristics and navigation behaviour analysis.

Vessel trajectory playback is widely used in the fields of vessel violation identification, accident investigation, evidence collection, etc. Figure 7 is an example of multi-vessel trajectory playback at the Yangtze River Estuary at a certain time on 28 April 2019. Double-clicking the yellow vessel AIS target activates the target. The static and dynamic information of the target can be viewed in the information window. The speed and progress of trajectory playback are controlled by a sliding bar, and the starting, pausing, and stopping of playback are controlled by buttons. During playback, the vessels' historical trajectories can be selectively displayed.

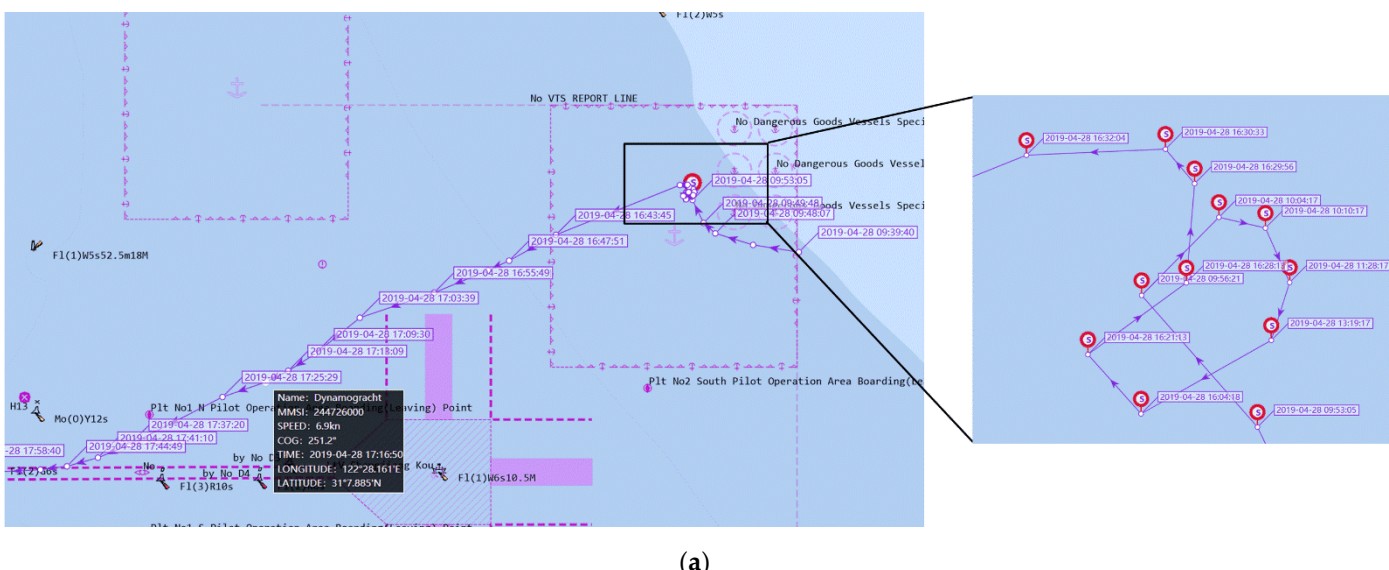

(**a**)

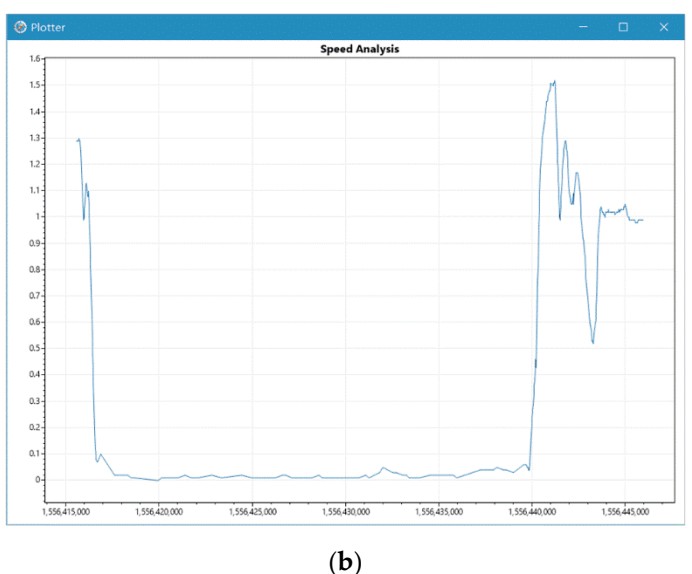

(**b**)

**Figure 6.** Single-vessel trajectory display: (**a**) the trajectory display of a single vessel; (**b**) speed analysis.

### 6.2. Vessel Trajectory Clustering

Vessel trajectory clustering can be used to find the motion patterns of vessels, analyze movement rules, and detect abnormal behaviour of vessels. In this paper, the FastDTW algorithm and the OPTICS algorithm are used to realize fast vessel trajectory clustering. Figure 8 shows the clustering results of vessel trajectories in the Yangtze River Estuary from 1 to 3 April 2019. Trajectories with the same flow direction and adjacent distances in the figure are clustered together, and different trajectory clusters are distinguished by color. The differences and similarities among traffic flows (clusters) can be intuitively found through color difference. By comparing the static and voyage reports of vessels, it can be seen that the draft of all vessels in traffic flows A and B in the figure is above 7 m, while the draft of all vessels in traffic flows C and D is below 7 m. Therefore, it can be concluded that "the north channel is the exclusive channel for vessels with larger draft, and the south channel is the exclusive channel for vessels with smaller draft", which is consistent with actual navigation. Through the study of the vessel trajectories in the traffic flow, the differences and similarities of the vessels in the traffic flow can be further found, such as the subtle differences in the vessel type and voyage time of the vessels in the cluster.

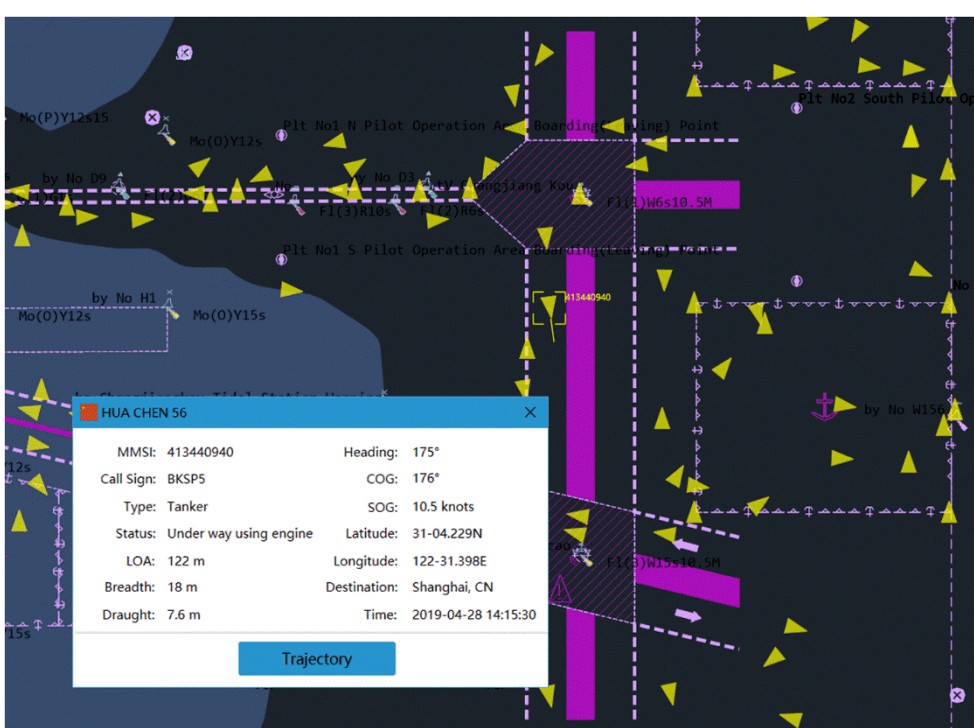

**Figure 7.** An example of multi-vessel trajectory playback in the Yangtze River Estuary at a certain time on 28 April 2019.

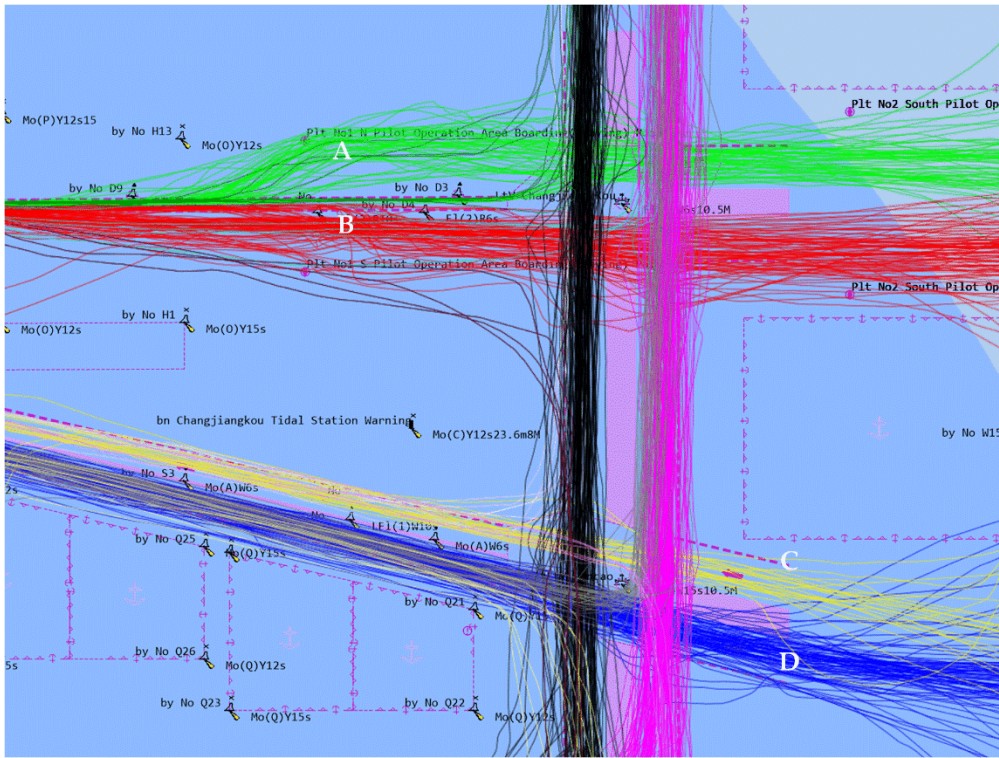

**Figure 8.** Clustering results of vessel trajectories in the Yangtze River Estuary from 1 to 3 April 2019.

*6.3. Vessel Heatmap*

This paper designs a vessel heatmap, uses color to reflect the data, and shows the distribution and density trends of vessels from the perspective of space and time. Figure 9 shows the vessel heatmap at a certain time on 10 April 2019. According to the principle of a heatmap, the darker the red color in the heatmap, the higher the density, so the density of vessels in areas A, B, C, and D in the figure is higher. Generally speaking, there are many vessels in the intersection waters, and the density of vessels is higher there. According to the ship routing system of the Yangtze River Estuary, it can be seen that areas B, C, and D in the figure are the intersection waters, which proves the effectiveness of the heatmap to a certain extent. By continuously observing the heatmap of the waters, it can be found that the density of vessels in areas B, C, and D changes greatly before and after high and low tides, while the density of vessels changes relatively little at other times. In navigation practice, vessels generally take the tide to enter and exit the estuary at high and low tide times, and the density of vessels in waters B, C, and D is relatively high at this time, which also proves the effectiveness of the heatmap to a certain extent.

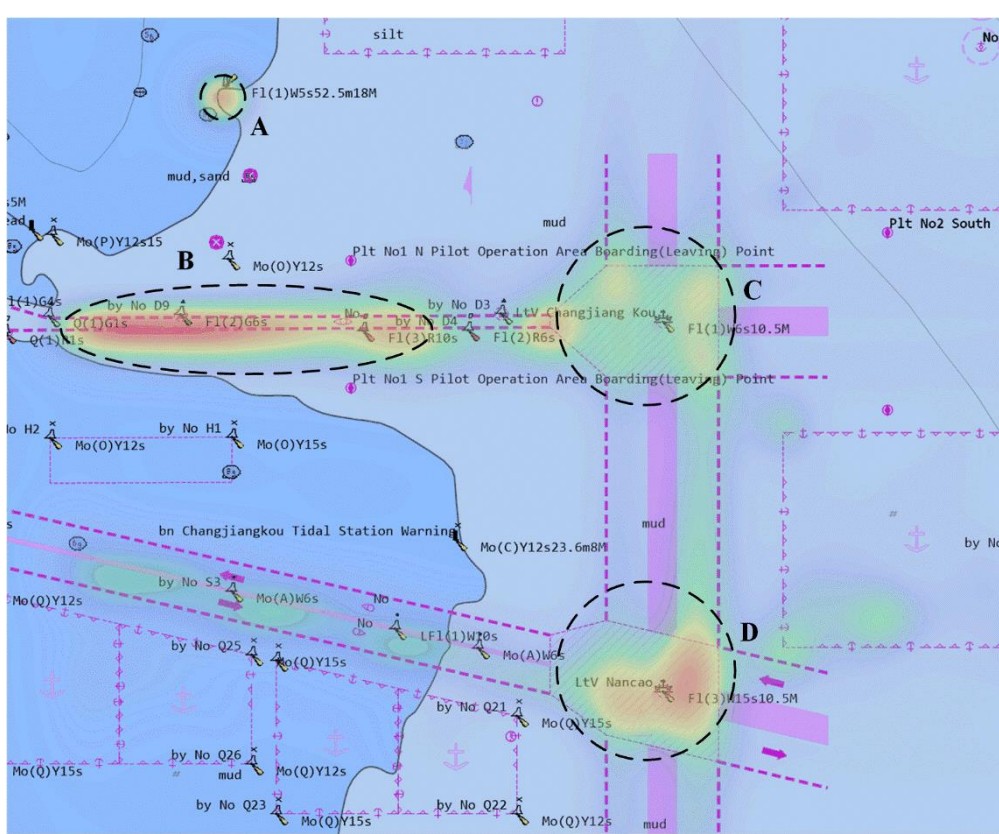

**Figure 9.** Heatmap of vessels in the Yangtze River Estuary at a certain time on 10 April 2019.

## 7. Conclusions

This paper designs a visual analysis system combining multiple views to explore and analyze vessel behaviour patterns. This paper completes the storage and acquisition of vessel trajectory data by data warehouse; uses a trajectory compression algorithm that considers the characteristics of vessel motion to preprocess the trajectory data; proposes an algorithm based on the principle of spatiotemporal density to detect stay points of vessels and label the vessel trajectory semantically; uses the FastDTW similarity measurement algorithm and the OPTICS clustering algorithm to achieve the rapid clustering of vessel trajectories; and combines the view of trajectory display and playback, the view of trajectory clustering, the view of vessel heatmap, etc., to display vessel trajectory data in multi-view

and multi-mode displays. An empirical case study of the Yangtze River Estuary was carried out using the proposed method, indicating that the system is practical and effective.

For the visual analysis of vessel trajectory data, the research in this paper still has some drawbacks. For example, when dealing with a large number of trajectories, the system efficiency is not high; some views are not aesthetically pleasing and some operations are not user-friendly enough; the types of views are not rich enough, and the data analysis is not in-depth; and there is a lack of consideration for vessels without AIS or equipped with AIS that is not used in a standard way.

Future research will be carried out considering the following directions: (1) using GPU to accelerate the calculation of the FastDTW distance measure to improve the efficiency of trajectory clustering; (2) adopting Model-View-ViewModel (MVVM) architecture to reduce system coupling and simplify the system operation process to improve the user experience; (3) combining other attributes of trajectory data and using more advanced visualization techniques (such as a stacked space–time cube or dynamic map) to build richer and in-depth analytic views; (4) attempting to visualize vessel behaviour based on vessel video surveillance images, shore-based surveillance radar images, LiDAR point clouds, and Synthetic Aperture Radar (SAR) images in the future.

**Author Contributions:** Ye Li led the writing of the manuscript, contributed to data analysis and research design; Hongxiang Ren supervised this study, contributed to funding acquisition, and serves as the corresponding author. All authors have read and agreed to the published version of the manuscript.

**Funding:** This research was funded by National Natural Science Foundation of China, grant number: 52071312 and 51939001.

**Institutional Review Board Statement:** Not applicable.

**Informed Consent Statement:** Not applicable.

**Data Availability Statement:** Data sharing is not applicable to this article.

**Acknowledgments:** The authors express thanks to anonymous reviewers for their constructive comments and advice.

**Conflicts of Interest:** The authors declare no conflict of interest.

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
