# Peer review of "Visual Analysis of Vessel Behaviour Based on Trajectory Data: A Case Study of the Yangtze River Estuary"

_ijgi, doi:10.3390/ijgi11040244_

Round 1

Reviewer 1 Report

The submitted paper deals with the development of an application for the visualisation and analysis of AIS maritime traffic data.
Overall, in my opinion, the paper is well structured, clearly written and illustrated and its bibliography is well selected and formatted.

On the form, my remarks concern a few points: the wording of certain sentences could be improved, the addition of a general flow chart would allow readers to better understand the structure of the application presented, the resolution of the illustrations should be increased to improve their readability.

In terms of content, the subject of the article is well introduced and the problematic well presented, but a few additions would improve its context. The state of the art in particular should be discussed more (see my comments directly in the article file).
The method is clearly explained but would be even clearer with the addition of a general flowchart.

Finally, in my opinion, the conclusion should be further developed by providing guidance on methods to address the limitations currently identified. The next steps of the project should be presented. Finally, the operational perspectives of the developed application could be recalled.

In conclusion, from my point of view as a generalist reviewer (and not a specialist in the processing of massive data), the article seems to me to be entirely worthy of being retained for publication, albeit with a few corrections of form and a few additions of substance.

Author Response

Response to Reviewer 1 Comments

Dear Reviewer 1:

Thank you for your letter and for the comments concerning our manuscript entitled “Visual Analysis of Vessel Behaviour Based on Trajectory Data” (ID: ijgi-1612843). Those comments are all valuable and very helpful for revising and improving our paper, as well as the important guiding significance to our researches. We have studied comments carefully and have made correction which we hope meet with approval. Revised portion are marked in the paper. The main corrections in the paper and the responds to the comments are as flowing:

Point 1: The wording of certain sentences could be improved, the addition of a general flow chart would allow readers to better understand the structure of the application presented, the resolution of the illustrations should be increased to improve their readability.

Response 1: Thank you for your careful work. We have carefully revised the article, revising sentences with wording or grammar problems. We have added a workflow chart in Section 2 to interpret the structure of the application. All illustrations in this paper have been replaced, ensuring that each image is of high resolution. Please see attachment for details.

Point 2: In terms of content, the subject of the article is well introduced and the problematic well presented, but a few additions would improve its context. The state of the art in particular should be discussed more (see my comments directly in the article file).

Response 2: Thank you for the valuable suggestions. We carefully checked every comment in this paper and revised the existing problems one by one. Especially for the modification of Section 1, we detail the contributions and limitations of the related work, and add some discussion related to the approach presented in this paper. Please see attachment for details.

Point 3: In my opinion, the conclusion should be further developed by providing guidance on methods to address the limitations currently identified. The next steps of the project should be presented. Finally, the operational perspectives of the developed application could be recalled.

Response 3: Thanks for your excellent suggestion, we have rephrase the conclusion of the article in the way you suggested. Please see attachment for details.

Reviewer 2 Report

This paper is fine and useful for safe navigation in maritime field, but i recommend the following points.

>The regulation of AIS defines in IMO. Add the regulation in the references. The reader understand well about AIS.

> The small vessel has NOT AIS. Add to explain on idea for non-AIS vessel.

> The used AIS data is only one day or a few day. That is NOT big data.  Show more than 1 year data if you are possible because the behavior depends time and season.

> The used AIS data is limited area, so add sub-title, ex. Case study of ....

Author Response

Response to Reviewer 2 Comments

Dear Reviewer 2:

Thank you for your letter and for the comments concerning our manuscript entitled “Visual Analysis of Vessel Behaviour Based on Trajectory Data” (ID: ijgi-1612843). Those comments are all valuable and very helpful for revising and improving our paper, as well as the important guiding significance to our researches. We have studied comments carefully and have made correction which we hope meet with approval. Revised portion are marked in the paper. The main corrections in the paper and the responds to the comments are as flowing:

Point 1: The regulation of AIS defines in IMO. Add the regulation in the references. The reader understand well about AIS.

Response 1: Thank you very much to point out the issue in our manuscript. We have added some references on AIS regulations and a detailed description of AIS data accuracy in the manuscript. For instance, the “International Maritime Organization (IMO). Resolution A.1106 (29): Revised Guidelines for The Onboard Operational Use of Shipborne Automatic Identification Systems (AIS); IMO: London, UK, 2015.” and the “International Telecommunication Union Radiocommunication Sector (ITU-R). Recommendation ITU-R M.1371-5: Technical Characteristics for an Automatic Identification System Using Time Division Multiple Access in the VHF Maritime Mobile Frequency Band; International Telecommunication Union Radiocommunication Sector (ITU-R): Geneva, Switzerland, 2014.”. Please refer to the attachment for details.

Point 2: The small vessel has NOT AIS. Add to explain on idea for non-AIS vessel.

Response 2: I have to say that this is a very very professional problem, and it is also a very difficult problem. The IMO requires AIS to be fitted aboard all ships of 300 gross tonnage and upwards engaged on international voyages, cargo ships of 500 gross tonnage and upwards not engaged on international voyages and all passenger ships irrespective of size. Therefore, some small ships are not required to install AIS. At the same time, there are also some ships with AIS that are not used properly. In view of this problem, we plan to try to visualize vessel behaviour based on ship video surveillance images, shore-based surveillance radar images, LiDAR point clouds and Synthetic Aperture Radar (SAR) images in the future (explain at the end of the paper). Please refer to the attachment for details.

Point 3: The used AIS data is only one day or a few days. That is NOT big data. Show more than 1 year data if you are possible because the behavior depends time and season.

Response 3: Thank you very much for your professional comments. We are aware of the problem, but for funding reasons, we only got the data for one month. This year, we plan to procure more up-to-date AIS data for research. Due to the large number of ships in the Yangtze River Estuary, in order to clearly display the trajectory data and better display effect, we only selected part of the data. And we have added descriptions of some algorithms in this paper to ensure that the algorithms are usable, and provided clearer illustrations.

Point 4: The used AIS data is limited area, so add sub-title, ex. Case study of .

Response 4: Thanks for your advice. We have added a subtitle to the title of this paper. Now the title of this paper is “Visual Analysis of Vessel Behaviour Based on Trajectory Data: A Case Study of the Yangtze River Estuary”.

Reviewer 3 Report

  1. AIS data are important, but they have a delay period, so it is not always possible have very high accuracy of the trajectory and to use them for operational purposes, as the authors state in section 3.1.1. Recommended point out it. 
  2. Based on AIS, it is difficult to accurately estimate the maneuverability of a ship, as the author’s state in Section 3.1.1, and it is therefore recommended that authors either specify accuracy limits or not claim that high accuracy results may be obtained.
  3. Table 1 shows part of the program, but not the algorithm, so it is recommended to provide a true algorithm and specify the incoming dimensions
  4. The first paragraphs of section 3.2.2 are week statements, it is recommended to either "strengthen" or abandon.
  5. The "scattering" of the ship 's trajectory referred to in 3.2.3, in particular when the ship is moored at the quay, is not the result of the ship' s motion but of the fluxing of the ship 's positioning system. Therefore, it is incorrect to plan the distance to the quay between ships on the basis of the indicated fluctuation, it is recommended to define it precisely.

Author Response

Response to Reviewer 3 Comments

Dear Reviewer 3:

Thank you for your letter and for the comments concerning our manuscript entitled “Visual Analysis of Vessel Behaviour Based on Trajectory Data” (ID: ijgi-1612843). Those comments are all valuable and very helpful for revising and improving our paper, as well as the important guiding significance to our researches. We have studied comments carefully and have made correction which we hope meet with approval. Revised portion are marked in the paper. The main corrections in the paper and the responds to the comments are as flowing:

Point 1: AIS data are important, but they have a delay period, so it is not always possible have very high accuracy of the trajectory and to use them for operational purposes, as the authors state in section 3.1.1. Recommended point out it.

Response 1: Thank you for the suggestion. As you said, AIS cannot always be highly accurate. The main function of AIS is ship identification and academic research, not real-time ship collision avoidance. To solve this problem, we add a table and a reference (International Telecommunication Union Radiocommunication Sector (ITU-R). Recommendation ITU-R M.1371-5: Technical Characteristics for an Automatic Identification System Using Time Division Multiple Access in the VHF Maritime Mobile Frequency Band; International Telecommunication Union Radiocommunication Sector (ITU-R): Geneva, Switzerland, 2014.) to illustrate the accuracy of the AIS data. And we add a detailed description of the limitations of AIS in Section 3.1.1. Please refer to the attachment for details.

Point 2: Based on AIS, it is difficult to accurately estimate the maneuverability of a ship, as the author’s state in Section 3.1.1, and it is therefore recommended that authors either specify accuracy limits or not claim that high accuracy results may be obtained.

Response 2: Thank you for your professional comments. Here, we would like to apologize for our impreciseness.We add a table to illustrate the accuracy of the AIS data and we rephrase the sentence. We also add a detailed description of the limitations of AIS in Section 3.1.1.

Point 3: Table 1 shows part of the program, but not the algorithm, so it is recommended to provide a true algorithm and specify the incoming dimensions.

Response 3: Thank you very much to point out the issue in our manuscript. In Section 3.2.1, we add a detailed description of the algorithm and specify the incoming dimensions. Considering the readability of the program, we keep the pseudocode in the paper. At the same time, we also provide real algorithm code written in Python language, and hope to get your suggestions for improvement.

from vincenty import vincenty
import pandas as pd
import matplotlib.pyplot as plt

def sbc(s, max_dist_threhold, max_spd_threhold):
    scopy = pd.DataFrame.copy(s, deep=True)
    s = scopy.reset_index(drop=True)
    if len(s) <= 2:
        return s
    else:
        is_halt = False
        e = 1
        while e < len(s) and not is_halt:
            i = 1
            while i < e and not is_halt:
                Δe = s.at[e, 't'] - s.at[0, 't']
                Δi = s.at[i, 't'] - s.at[0, 't']

                xp = s.at[0, 'lat'] + (s.at[e, 'lat'] - s.at[0, 'lat']) * Δi / Δe
                yp = s.at[0, 'long'] + (s.at[e, 'long'] - s.at[0, 'long']) * Δi / Δe
                ptp = (xp, yp)
                pta = (s.at[i, 'lat'], s.at[i, 'long'])

                Vi_1 = vincenty(pta, (s.at[i - 1, 'lat'], s.at[i - 1, 'long'])) / (
                        s.at[i, 't'] - s.at[i - 1, 't']) * 1000 / 1852.25 * 3600
                Vi = vincenty((s.at[i + 1, 'lat'], s.at[i + 1, 'long']), pta) / (
                        s.at[i + 1, 't'] - s.at[i, 't']) * 1000 / 1852.25 * 3600
                if vincenty(pta, ptp) * 1000 / 1852.25 > max_dist_threhold or abs(Vi_1 - Vi) > max_spd_threhold:
                    is_halt = True
                else:
                    i = i + 1
            if is_halt:
                return pd.concat([s[0:1], sbc(s[i:], max_dist_threhold, max_spd_threhold)], ignore_index=True)
            e = e + 1
        if not is_halt:
            return s.loc[[0, len(s) - 1]]

if __name__ == '__main__':
    data = pd.read_csv('out.csv')
    data.rename(columns={'DRLONGITUDE': 'long', 'DRLATITUDE': 'lat', 'DRGPSTIME': 't'}, inplace=True)

    data['long'] = data['long'].apply(lambda x: x * (1 / 600000))  # ° as unit
    data['lat'] = data['lat'].apply(lambda x: x * (1 / 600000))

    simpletraj = sbc(data, 0.1, 0.5)

    simpletraj.to_csv('myfile.csv')

    # 2d
    fig, ax = plt.subplots()
    ax.plot(data['long'], data['lat'], marker='*', alpha=0.5, linestyle='-.', label='actual')
    ax.plot(simpletraj['long'], simpletraj['lat'], marker='o', label='simple')
    ax.legend()
    plt.show()

Point 4: The first paragraphs of section 3.2.2 are week statements, it is recommended to either "strengthen" or abandon.

Response 4: Thank you for your careful work. Considering your suggestion, we have abandon the statements and rephrase the first paragraph of Section 3.2.2.

Point 5: The "scattering" of the ship 's trajectory referred to in 3.2.3, in particular when the ship is moored at the quay, is not the result of the ship' s motion but of the fluxing of the ship 's positioning system. Therefore, it is incorrect to plan the distance to the quay between ships on the basis of the indicated fluctuation, it is recommended to define it precisely.

Response 5: Thank you very much for your comment. Unfortunately, we didn't understand all of your comments. But your comments give us great enlightenment, especially this sentence ”in particular when the ship is moored at the quay, is not the result of the ship's motion but of the fluxing of the ship's positioning system”. In the stay point detection algorithm proposed in Section 3.2.2 of this paper, the algorithm results may be affected by 'GPS/GNSS drift'. After statistical calculation, we find that the influence of 'GPS/GNSS drift' on ship position in berthing is very small compared with that caused by ship motion in anchoring. Therefore, it is more appropriate to use the standard deviation of the speed or the standard deviation of the position to judge the type of the stay point. At the same time, we also revised the paper. Please refer to the attachment for details.

Round 2

Reviewer 3 Report

  1. Table 1 recommended to provide more explanation of the incoming dimensions

Author Response

Response to Reviewer 3 Comments

Dear Reviewer 3:

Thank you for your letter and for the comment concerning our manuscript entitled “Visual Analysis of Vessel Behaviour Based on Trajectory Data” (ID: ijgi-1612843). This comment is very valuable and very helpful for revising and improving our paper, as well as the important guiding significance to our researches. We have studied the comment carefully and have made correction which we hope meet with approval. Revised portion are marked in the paper. The main corrections in the paper and the responds to the comment are as flowing:

Point 1: Table 1 recommended to provide more explanation of the incoming dimensions.

Response 1: Thank you for your valuable suggestion. For Table 1, we have adjusted the table and enriched the contents of it. For Table 2, we have added some explanations of the incoming parameters and some code annotations. Please see attachment for details.
